# Analysis of Evaluation Dimensions of Public Service Motivation of Chinese College Students—Qualitative Study Based on Grounded Theory

**DOI:** 10.3390/ijerph192215084

**Published:** 2022-11-16

**Authors:** Hongming Zhang, Qingya Zhang, Guoliang Huang, Jin Ke, Ni Zhao, Wanting Huang, Jun Zhang

**Affiliations:** 1School of Economics and Management, South China Normal University, Guangzhou 510631, China; 2School of Marxism, South China Normal University, Guangzhou 510631, China

**Keywords:** public service motivation, college students, collectivist tendency, self-improvement

## Abstract

Public service motivation (PSM) represents an individual’s predisposition to respond to motives grounded primarily or uniquely in public institutions, and it is an individual characteristic that reflects the attributes of the public sector. The concept was first introduced by James Perry, who identified four-dimensions to measure PSM, namely, attraction to policy making, commitment to the public interest, compassion, and self-sacrifice. Public service motivation changes over time, and differences in culture and systems in different countries can lead to differences in the measurement dimensions of PSM. The dimensions of PSM measurement in Asian countries are different from those in Western countries, and whether the regional applicability and population applicability of PSM can be expanded is a question worth investigating. From a new perspective, this study takes Chinese college students as the research object, using one-on-one interviews based on grounded theory. Besides the four dimensions mentioned above, it was found that the two additional dimensions of a collectivist tendency and self-improvement were added, and the specific connotations of each dimension were changed somewhat. A collectivist tendency accords with the culture of East Asian countries, and self-improvement is our unique finding among college students. This proves that the motivation for public service can rise not only from altruism but also as a result of individuals seeking self-development and value realization.

## 1. Introduction

Public service motivation (PSM) represents a person’s pro-social motivation to help others and serve the public interest by providing public service [1], and it is a set of beliefs, values, and attitudes that transcend self-interest and organizational interests, which relate to the larger political entity interests of the larger political entity and motivate individuals to act accordingly when appropriate. There have been studies on public service motivation, mainly on conceptual definition [2], sources of motivation [3,4], job impact [5,6], scale of development [7], and differences in influencing factors across contexts [8]. As for the measurement of PSM, since the emergence of the concept of PSM there have been numerous debates on its cross-cultural explanatory power, and cultural differences not only affect the level of PSM [9] but also have the potential to influence the intrinsic structure of PSM, which in turn affects its reliability and validity of measurement in different cultural contexts. In recent years, the localization of domestic public service motivation research has been developing rapidly. More experts have realized the significance of carrying out public service motivation research based on local conditions. It can not only make public service motivation play a role in practice but also repair and supplement it in the Chinese context [10]. Perry earlier combined Kononk’s three-category division for motivation [11] with PSM and designed 40 questions of PSM based on the historical context of the United States, forming four core dimensions including attraction to policy making, commitment to the public interest, compassion, and self-sacrifice [12]. Leisink and Steijn [13] applied a streamlined version of Perry’s scale to the Dutch public sector, and the study verified the applicability of Perry’s four-factor theory in the Netherlands. Vandenabeele et al. [14], however, based on a comparative study between the UK and Germany, found that bureaucratic traditions, mechanisms of service delivery, and perceptions about equality influenced the applicability of Perry’s four dimensions, and the four dimensions showed significant differences across countries. Similar studies have discussed the applicability of PSM in France [15], Korea [16,17], and China [18], but few studies have focused on the dimensions of PSM measurement in college student populations.

Scholars have conducted rich research on both the antecedents and consequences of PSM [1,19,20]. PSM is influenced by personal characteristics such as family education, military service, religious beliefs, volunteer activities, and occupation [8]. Public service motivation changes over time, with an increasing number of studies focusing on longitudinal changes in PSM, i.e., pre-entry differences in PSM, and research objects expanding from civil servants to college student populations, such as the influence of vocational studies [21], cross-educational experiences [22], and community philanthropy activities [23] on college students’ PSM. Moreover, differences in culture and systems in different countries may also lead to differences in the measurement dimensions of PSM. Cross-cultural research has been an important area of PSM research, and the dimensions in Asian countries are to some extent different from those in Europe and the United States [17]. China is influenced by Confucian culture and has a special cultural background and social system. Yang (2021) [18] explored the Chinese dimensions of PSM for civil servants, adding two dimensions of social reputation and moral adherence based on the Perry four-dimensional scale. According to Perry (2000) [24], three main factors influence PSM. The first is the socio-historical context, which consists of the individual’s education, socialization, and life events; the second is the individual’s motivational context, which includes the work environment, organizational incentives, and job characteristics, as well as the beliefs, values, and ideologies of the organization in which the individual works; and the third is personal characteristics, such as abilities and competencies, as well as the values and identities that constitute self-awareness. Culture and experience influence the dimensions of PSM. Are the dimensions of PSM of college students who have not yet entered the workforce different from previous research? College students’ PSM may have a long-term and positive impact on later achievement and well-being [25]. Therefore, it is helpful and important to study the measurement dimensions of PSM among college students and compare the similarities and differences with those of civil servants.

College students are the driving force and reserve force of the country’s high-quality development. People with high PSM perform better at work and in life [26], so the study of PSM in the context of higher education is important for the development of excellent college students. This study takes Chinese college students as the research objects and conducts qualitative research. In this paper, we used one-on-one telephone interviews and under the guidance of rooting theory summarized the 210,000 words of interview materials from 30 interviewees using NVivo software. This study constructed six measurement dimensions of PSM We found that Perry’s four dimensions of PSM can be generalized among Chinese college students, but in addition to these four dimensions (attraction to policy making, commitment to public interest, compassion, and self-sacrifice), two additional dimensions of collectivist tendency and self-improvement were added, and the specific connotation of each dimension has changed somewhat.

The contributions of this research are as follows: Firstly, it enriches the theory of PSM. While most previous studies have focused on the PSM of civil servants, college students, as an important force for social development, have their characteristics and are a group that cannot be ignored because of their large number. This study explores the factors influencing PSM among college students in China, develops Chinese measurement dimensions of PSM, and broadens the scope of application of PSM. Secondly, it shows the differences in research on PSM between developed and developing countries. Kim (2009) [17] extended the Perry scale to Korea and modified the original scale. In the Asian cultural context, this paper constructs six measurement dimensions of PSM, the results of which differ from Kim’s study, indicating that there are still differences in PSM measurement among Asian countries. Thirdly, new dimensions of the PSM scale were developed. Based on the PSM scale created by Perry, collectivist tendencies and self-improvement were added, while the specific items of this dimension of attraction to policy making changed considerably to fit the college student context. The discovery of the new dimensions is significant because they are unique to Chinese college students’ PSM and more accurately reflect the antecedents of Chinese college students’ PSM. The new dimensions not only reflect the characteristics of Chinese culture and education but also demonstrate that the measurement dimensions of PSM have acquired new meanings among Generation Z, and that PSM, in addition to being altruistic, may also imply the purpose that individuals seek self-development and value-realization.

## 2. Literature Review

### 2.1. Understanding of Public Service Motivation

As the study of public service progresses, the term public service no longer refers only to the service of those who work in the public sector but has expanded to include an attitude, a sense of responsibility, and even a sense of public morality. Public service motivation was first introduced by Perry and Wise in their paper “The Motivational Bases of Public Service” and refers to the psychological tendency of individuals to respond sensitively to important or specific goals of the public sector. Since then, numerous different understandings of public service motivation have emerged, which can be summarized into four perspectives. The first perspective views public service motivation as a special motivation of public sector practitioners that is different from that of private sector practitioners, seeking intrinsic meaning and value in their work [27,28], while the second perspective defines public service motivation simply as altruistic motivation [5]. The third perspective builds on the second perspective by expanding public service motivation to emphasize concern for the overall good of society rather than individual welfare [2], and the fourth perspective views public service motivation as a holistic approach that encompasses social history, organizational context, rationality and emotion, norms, and sacrifice [10,28]. Our study agrees with the fourth perspective that public service motivation is closely related to personality traits, social experiences, and family education [8], so it is impossible to generalize the concept from a single dimension, and multiple dimensions must be established to evaluate public service motivation.

Perry and Wise (1990) [11] distinguished three types of motivations for public service, which are rational, normative, and affective motives. Rational motivation refers to employees’ desires to participate in good public decision-making because they perceive it as a social responsibility or because it reinforces their image of self-importance and self-esteem; rational motivation recognizes the existence of a personal utility maximization component in public service. Normative motivation refers to the altruistic desire to serve the public good or to serve the state to promote social justice, and personal values play an important role in public service motivation. Emotional motivation includes identification with public service, emotional connection with service users, compassion, and self-sacrificing contribution based on one’s benevolence, patriotism, or heroism that makes service providers feel happy in sacrificing and giving to others, which triggers the individual’s willingness to sacrifice for others.

To better measure PSM and conduct more empirical studies, scholars have focused on the development and refinement of PSM scales. Perry (1996) [12] used questionnaires and reliability and validity tests to categorize PSM into four dimensions, namely, attraction to policy making, commitment to the public interest, compassion, and self-sacrifice, with a total of 24 question items. On this basis, Perry (1997) [29] proposed that personal experiences affect PSM, such as childhood experiences, religion, and occupation. Kim (2009) [17] considered the dimensions of PSM measurement in the Korean context, revised the original Perry scale to a 4-dimensional, 12-item scale, and modified the APM dimensions to better measure PSM, and subsequent studies on PSM have been based on three factors (based on Kim’s study with the APM dimensions removed) or four factors with minor modifications [13,30,31].

### 2.2. PSM of Different Study Subjects

The PSM measurement model originated in the United States, and most of the initial research on measurement was conducted in the United States [4,5,12], but studies based on different national contexts have enriched the scope and dimensions of PSM research. Taylor (2007) [32] surveyed 203 employees of public agencies in Australia and found that rational motivation had significant individual effects on job satisfaction, organizational commitment, and work motivation, while bureaucratic red tape led to a decline in PSM. Yang and Yang (2021) [18] developed a public service motivation scale suitable for China by considering the cultural adaptability of public service motivation, adding two dimensions (social recognition and fame, and moral persistence) to Perry’s original scale, while there were some differences in the presentation of the original dimensions. Liu et al. (2022) [27], based on Perry’s public service motivation theory, measured a village cadre’s public service motivation and the effectiveness of rural living environmental governance. Liu et al also explored the structure of public service motivation of grassroots civil servants in the Chinese context with Perry’s original scale [33]

PSM is associated with organizational citizenship behavior [34], performance, job satisfaction [35,36], goal clarity [37] organizational commitment, and other factors. Higher PSM not only increases the likelihood of working for the government [38], but it also increases job satisfaction and performance, and it reduces turnover among federal employees [39]. It also enriches multidimensional measures of organizational performance [40], improves organizational citizenship behavior [41], and encourages the improvement of the quality of their services. However, this finding varies by country, gender [42], religion, and family socialization [11,29], as well as between genders. Lee (2005) [30], using a 24-item scale, found that among Korean public officials, the “Attraction to Public Policy Making” dimension does not affect performance levels, but the other three components (commitment to the public interest, compassion, and self-sacrifice) affect performance levels. Parola et al. (2019) [43] found that in Asian Confucian countries (i.e., China, Taiwan, and Korea), PSM was higher for men, while in the United States, United Kingdom, and Australian countries, PSM was higher for women.

### 2.3. Research on PSM among College Students

The antecedents and consequences of PSM in this group of college students, who are the main group that will enter the public sector workforce, have also received more attention. Community service [23], theological reading [44], and higher college education [21] can increase PSM among college students. Lundy et al. (2018) [22] found that interprofessional educational experiences strengthen college students’ public service motivations, particularly in terms of public interest commitment and compassion. Miller et al. (2022) [45] longitudinally studied the evolution of PSM in a cohort of Israeli students who gradually entered the job market and found that individuals with strong collectivism and an academic background in the core public service had higher PSM over time. Ngaruiya et al. (2014) [46] found that ROTC (Reserve Officer Training Corps) participants had higher PSM based on the Kim Revised Scale. ROTC trainees had higher PSM than the general undergraduate population, and their enlisted institutional motivators (the institutional motivators) were positively correlated with the rational, normative, and affective dimensions of PSM. Van et al. (2017) [47] used the Big Five personality to explain PSM, and public service motivation was influenced by psychological factors. The emotional motivators of PSM—compassion (COM) and self-sacrifice (SS)—are influenced by the personality traits of honesty, humility, emotionality, and agreeableness positively and by conscientiousness negatively. In contrast, non-emotional PSM—attraction to policy making (APM) and commitment to the public interest (CPI)—was positively associated with the openness to experience trait.

Students with higher levels of PSM are more likely to engage in charitable giving and volunteering [48], and in particular, the self-sacrifice dimension can significantly increase willingness to volunteer. Using Chinese college students during the COVID-19 pandemic, Geng et al. (2022) [49] found that undergraduate students who were more concerned about the public interest and less likely to prefer private interests had the highest overall voluntary service rate, and individuals with altruistic motivations and compassion were more likely to volunteer out of concern for the public interest. PSM also affects political orientation, as Seider et al. (2011) [44] showed that participation in community service learning increased college students’ sense of civic responsibility and commitment to civic behavior, and that college students who engaged in service learning after six years of college were more likely to be politically engaged than their peers who did not engage in service during college.

PSM also affects employment intentions and job satisfaction. College students with a high level of public service ethos were more likely to pursue public sector careers [50] and public sector job seekers were more likely to value “high prestige and social status” and stability than private sector job seekers. Moreover, the willingness to choose a public job varies by gender and family environment. However, some studies show no effect of PSM on employment intentions. Gossett et al. (2016) [51] studied the motivation of management students in Botswana, an African country, to participate in the public sector, but did not find a relationship between PSM and participation in public sector jobs. Possible explanations are that (1) college students in some countries have different motivations to seek employment because of the institutional system in their country, and they choose public sector jobs for other reasons. Ko and Jun (2015) [52] found that Asian students choose to go to work in the public sector for security and stability, but seeking opportunities to benefit society is also one of the factors. (2) Whether the Perry and Kim scale can be generalized to college student populations in different countries is an issue; Lee and Choi (2016) [53] used pro-social behavior as a proxy for the Kim modified scale to measure PSM, and the results of the study were essentially the same as those obtained using the Kim modified scale to measure PSM, which indicates that the PSM scale focuses on different dimensions in different countries, and in the Korean college student population, pro-social behavior was consistent with PSM as measured by self-sacrifice, compassion, and commitment to the public interest.

## 3. Research Process

### 3.1. Introduction of the Research Object

The college years are a particularly important period because they are characterized by increasing independence and responsibility [49]. From a life course perspective, PSM in college students may have long-term and positive effects on later achievement and well-being [25]. Studying the evaluative dimensions of PSM among college students and comparing the similarities and differences between them and those of the civil service population helps us to better identify the heterogeneity of PSM among college students and facilitate more extensive research in the future.

In terms of social culture, East Asian countries are generally characterized by high power distance [54], a collectivist tendency [55], and an emphasis on internal harmony and uncertainty avoidance. Under the influence of traditional Confucianism, China has always had altruistic values such as “benevolence and virtue” and “kindness and charity”. However, during the feudal period, there was a lack of equal relationship between the government and the people, and more emphasis was placed on “ruling” than “serving”. The contemporary Chinese government emphasizes the mission of “serving the people wholeheartedly,” but in the actual governance process, the behavior of public officials is largely regulated by the vertical hierarchy [56], which makes the measurement of public service motives more complex. Humility and euphemisms are characteristics of East Asian cultures, which requires scales to be more appropriate for Chinese contexts, such as the expression “politics is dirty” in the “attractiveness of policy making” dimension of the Perry scale, which may be difficult to measure accurately in Chinese cultural contexts.

In terms of student characteristics, on the one hand, modern Chinese education is full of East–West fusion, and Chinese students have long been guided to focus on harmony, pragmatism, reality, dedication, and collective honor. In ideological and political education, emphasis is placed on the need for youth to struggle courageously, establish high social ideals, and unite personal ideals with social ideals. Especially in the public sector, it also emphasizes that when individual interests conflict with collective interests, the collective interests should come first. Therefore, the culture of collectivism, patriotism, and self-sacrifice has long been deeply rooted in Chinese ideological and political education. However, at the same time, universities also focus on enhancing students’ creativity and expanding their practical skills, and the pursuit of truth and equal communication allows students to gradually break with authority. On the other hand, the needs and preferences of the younger generation have changed, with millennials (the generation reaching adulthood after entering the 21st century) making “balancing personal and professional life, continuing education, and contributing to society their primary career goals,” and the fusion of Western cultures and the fierce competition of the market economy making students more concerned with self-improvement. Students who are highly motivated by the public service tend to have higher expectations of themselves and hope to improve themselves while contributing to the public good through more ways.

In summary, we believe that to study young students’ motivations for public service, it is necessary to deeply understand young students’ thoughts and explore accurate evaluation dimensions from young students’ contexts.

### 3.2. Research Methods Based on Grounded Theory

In this paper, we used a qualitative research method to summarize the dimensions of PSM evaluation of college students through one-on-one interviews, and we used NVivo software to generalize the PSM evaluation dimensions based on grounded theory. NVivo software enabled us to classify, code, and analyze 210,000 Chinese characters in interview transcripts. Grounded theory is a bottom-up approach to construct theory proposed by Glaser and Strauss [57]. Grounded theory does not make assumptions before research, but extracts new concepts and ideas from empirical facts, supported by empirical evidence. It relies on three levels of coding to construct theory: open coding, axial coding, and selective coding. After collecting a large amount of primary material, a systematic procedure needs to be followed, including recording, analysis, coding, and excerpting until the final report is formed. In the open coding process, we carefully read the interview materials, marked key words and key sentences, and gave it an initial concept or label, and then we organized these concepts or labels through open coding to form categories, and then we formed the key concepts through axial coding. Finally, the key concepts were unified to represent PSM.

While previous PSM measurement dimensions are sufficiently mature, they lack applicability for the subject of this paper and cannot reflect the dimensions unique to college students, so we had to explore the dimensions that were more appropriate for college students. We had to draw conclusions from observations and interactions with the objects. Guided by grounded theory, qualitative research methods, such as observation and interview, are classical. We used the interview method, which is the most direct method to acquire the most primary materials. To accurately grasp college students’ understandings of public service motivation and to avoid conceptual guidance, one-on-one in-depth interviews were conducted. Considering the interviewees’ cognition and experiences, the interview questions included open questions and structured questions: “Do your friends and classmates have any behavior that reflects high public service motivation? Give at least one example and why you think they do so. What characteristics do they have?”, or some reverse questions, such as “You must have seen people with low public service motivation. What kind of people do you think to have low public service motivation?”. After all the open questions, we asked structured questions such as “How do you think volunteering is related to public service motivation?”, and “How do you think sacrificing one’s interests for the benefit of more people is related to public service motivation?”, etc., to enrich the definition of public service motivation.

### 3.3. Sampling Interview Process

First of all, in the initial stage of material collection, we purposefully selected a sufficiently typical sample of 20 students (11 male and 9 female) from different levels of universities in China, including “Project 985” institutions, “Project 211” institutions (i.e., National Key Universities or World-Class Universities), general undergraduate universities, private universities, and higher vocational colleges. Second, to reduce the possibility of self-disguise of the interviewees, this study adopted a “double-blind” design for all interviews, i.e., except for the project instructor, who did not participate in the interviews, both the interviewer and the interviewee did not know each other’s specific identity information, such as their school, whether they held student positions in the school, and their names, and both parties were only addressed by their last names in the interview. In addition, the interviewees were sophomores or juniors, including student leaders and non-student leaders, because we needed the interviewees to have a general understanding of PSM, and we wanted to learn more about college students’ perceptions of PSM from more perspectives. Overall, the sample of this study was typical in terms of school level, gender, and identity.

The interviews were conducted by online calls. The purpose of the interview was explained to the interviewees before the interview, and the confidentiality of the interview content was ensured. The interviews were recorded with the consent of the interviewees, and each interview lasted more than 10 min. Finally, we obtained 160,000 words of original interview materials. In the follow-up saturation test, we interviewed 10 different college students and obtained another 50,000 words of interview transcripts.

### 3.4. Interview Materials Coding

After preliminary coding, we were able to obtain 696 initial concepts or labels of motivation for public service and then organize these concepts or labels through axial coding. For example, we categorized “concern about policies and proactively defend students’ rights “ as “Proactive attention to school policies, activities, and student rights”; we categorized “feeling happy when helping others” as “have a strong sense of helping people”. The descriptions of “thinking about how to serve fellow students more conveniently and sacrificing one’s own time” were classified as “willingness to sacrifice personal interests to help others”; the descriptions of “taking responsibility with determination and feeling obligated to do so” were classified as “responsible and committed to work”. The interviewees also talked about the qualities of people with high motivation for public service, and we tentatively coded “gratitude, patience, hard work, kindness, commitment, and perseverance” as “personal qualities”. It is noteworthy that although the motivation of “award and merit” is egoism, most of the interviewees thought that this egoism did not prevent them from serving the people. Even if the public service is driven by egoism partially, the level of public services is not discounted. Therefore, we believe that egoism is one of the motivations for public service.

After initial coding, we concluded 696 labels, 61 categories, and 10 key concepts. Subsequently, for the initial coding of the interview content, we tried to correct it with more written, scientific, and professional coding. The process roughly went through the following steps: (1) comparing the categories and key concepts with Perry’s four dimensions of PSM and Yang and Yang’s (2021) [18] Chinese dimensions of PSM and integrating them into 49 open codes and 6 axial codes; (2) discussing with experts in ideological education and public administration to obtain more precise coding of these labels; (3) a saturation test, i.e., adding 10 interviewees from different colleges and observing whether new codes appear in order to ensure that the codes are tending to be saturated; (4) organizing the codes into a questionnaire and conducting a pilot survey of 340 students, again revising the expressions of each nodal statement to make it easier for interviewees to understand and enhance the accuracy of each dimension; (5) simplifying and merging—finally conducting 6 axial codes and 20 open codes for evaluating college students’ motivations for public service, with a total of 592 labels.

## 4. Coding Results Induction Analysis

In the open coding process, the paper suspended the personal “inclination” or “foresight”, summarized the “local concepts” from the primary sources, and formed 20 nodes. The axial coding linked the 20 nodes in the open coding to form six categories. The generalized dimensions are shown in Figure 1, and the nodes are shown in Table 1. The first four items were consistent with Perry’s original dimensions, while the latter two items were new to this study.

We abstracted 592 labels from the initial interview materials, then grouped these labels into 20 categories, i.e., open coding, and then grouped these 20 codes into 6 axial codes, with the number following every code representing the number of labels marked.

### 4.1. Attraction to Policy Making

Attraction to policy making, as one of the rational motives for public service motivation, refers to the desire of individuals to contribute to the society or community in which they live by participating in public policy making. People are willing to engage in public service because of the opportunity to participate in the formulation of public policy. Kelman [58] links the desire of individuals to participate in the formulation of good public policy to the norm of public spirit, and those who participate in public policy formulation may be serving the interests of society while satisfying their personal needs. This attraction is even more evident in Chinese society where the public sector has a great influence on society. In the college context, “paying attention to school policies and advocating for them” is one of the hallmarks of high public service motivation, which stems from the desire to contribute to fellow students and their organizations. In the interviews, most of the interviewees said that people with high PSM “pay attention to school policies and students’ rights”, “learn about national events and public events actively”, and “give some advice initiatively, as shown in Table 2. In the interviews, we distinguished the difference between “knowing more about public policies because of job responsibilities” and “actively learning about policies and actively promoting them”, and we only used the latter as the basis of coding.

Unlike civil service positions, college students provide services mainly through serving as student leaders, participating in clubs or voluntary organizations, and their political aspirations are weaker than those of social workers, and in addition to understanding national events and current affairs, more students are more concerned about school policies and student rights. In the campus environment, students hope to collect public opinions and improve their service through the open platform of the school or the platform they actively obtain due to the convenience of their positions, which is not only the exercise of their abilities but also the embodiment of public service spirit.

### 4.2. Commitment to the Public Interest

Commitment to the public interest is a normative motivation, which refers to the values-based motivation of individuals, such as social equality, the welfare of future generations, responsibility, and morality. Under the influence of Confucianism, every Chinese takes public interest into account since youth. Especially among college students, those with high public service motivation will think that it is their mission and obligation to serve everyone because of their status as party members, student cadres, or just highly educated people, and that it is their duty to take on responsibility in the organization.

The aspects involved in the commitment to the public interest in the Chinese college student context are richer than those in the Perry’s dimensions, as shown in Table 3. Firstly, those who are highly motivated by public service will actively engage in practical activities to serve the public and create value for others. Some people’s value pursuit is to excel in their professional field, while some people’s value pursuit is to take the initiative for public service and bring happiness to others. Secondly, the biggest difference between public service provided by students in universities and civil servants is that there are no material benefits (such as salary) and formal constraints (such as performance appraisal and departmental rules and regulations). Students provide public service relying on informal constraints such as responsibility and commitment: “*do something not only when asked to do it, but with enthusiasm*”, “*As a student leader, you should take the initiative to take responsibility for the mistakes of your ministry members instead of getting away with it*.” Thirdly, if a person works for personal gains, he or she is bound to be lazy in team work, while taking the initiative to undertake tasks in the public interest is a reflection of high PSM.

### 4.3. Compassion

Empathy is a category of emotional motivation and refers to an individual’s emotional commitment to the interests of others out of identity. Compassion requires taking the initiative to find others’ difficulties in daily life and giving a favor. During the interviews, we asked the interviewees about their perceptions of risking their lives to help others, and they said that these people have a high PSM. When others need help in public places, people with high PSM do not think “maybe others will come to help them”, but they are already doing it while thinking “I will go up and help them”. In addition, people with high PSM also choose to work on the front lines after graduation. The sub-nodes of “compassion” and examples are shown in Table 4.

### 4.4. Self-Sacrifice

Self-sacrifice is a core component of PSM and has altruistic, pro-social attributes [16], i.e., a greater willingness to give of oneself for the public good [59]. The sub-nodes of “self-sacrifice” and examples are shown in Table 5.

When performing student work at school, students who are highly motivated by public service will volunteer their time and energy to contribute to the public good. As student leaders, they are happy to spend more time solving the problem in order to make life easier for others. Although some students participate in volunteer activities for awards and merit, students with high PSM are willing to take the initiative to participate in volunteer activities that are unpaid and have no awards.

### 4.5. Collectivist Tendency

A collectivist tendency is one of the new dimensions found in this study and can be understood in terms of both affective and normative motives. This new dimension reflects the differences between Eastern and Western cultures. From the perspective of affective motives, in an environment with a strong collectivist culture, individual values are always linked to social values, and the sense of belonging and honor to the collective makes people more motivated by collective interests when providing public services. Firstly, in Chinese culture, individual behavior often represents collective behavior, and individual honor and shame are related to collective honor and shame. Under the influence of this social atmosphere, public service motivation is easily stimulated. In addition, the propagation of events of selfless devotion to the motherland and the collective will inspire college students to the practice of sacrificing for the collective. Most of the interviewees pointed out that civilian heroes who are willing to sacrifice selflessly for the greatest public interest are worthy of admiration and will be infected by similar news. Secondly, in campus life, the sense of collective belonging and collective honor is one of the important emotions that college students pursue on campus. Many people gain a sense of collective belonging by joining club organizations, and a strong sense of collective honor makes them more willing to give selflessly for the organization, and people with a sense of collective honor are willing to unite others, so it is easier for them to carry out student work.

From the perspective of normative motives, putting the collective interest first in collective work is the result of the ideological education that students have received over the years, and there is no material reward for carrying out student work but more on the spirit of dedication to the collective. This shows that college students regard it as an honorable thing to contribute to the collective. Compared with social work positions, being a student cadre is more a voluntary action, and no one makes specific requirements for the work of student cadres, so to do a good job in this position, one should have the spirit of willingness to contribute to the collective. The sub-nodes of “collectivist tendency” and examples are shown in Table 6.

### 4.6. Self-Improvement

Self-improvement is another new dimension found in this study, stemming from the individual’s pursuit of his or her comprehensive qualities, and should be classified as a rational motive. Early motivation theorists considered prestige as an incentive stemming from organizational size and growth, and according to Maslow’s hierarchy of needs theory, human needs are divided into five levels—physiological needs, security needs, belonging and love needs, respect needs, and self-actualization needs; and self-improvement covers belonging and love needs, respect needs, and self-actualization needs. Among the groups of college students interviewed, many subjects indicated that they joined student work to make friends, improve their communication and work skills, and gain a sense of self-fulfillment in serving the public, as shown in Table 7. To summarize the performance of PSM, interviewees mentioned expanding interpersonal relationships 23 times, enhancing personal overall quality 41 times, sense of accomplishment and honor 19 times, and achieving self-worth 15 times.

In university life, active participation in public service is regarded as an important channel to expand interpersonal relationships. If students want to gain valuable experience and training opportunities, they will certainly participate in public service. In addition to the belief that public service can create value for others, people with high PSM will also recognize their value when providing services.

### 4.7. Evidence from Word Frequency Statistics

In the process of coding analysis, we made word frequency statistics to further support the conclusions above, as shown in Table 8. It is interesting that “student union” is mentioned 377 times, and “volunteer” is mentioned 222 times. Most undergraduates agree that the main way to realize public service is to join the student union or volunteer. There are also some high-frequency words supporting our dimensions, and these words confirmed the rationality of induced dimensions in Table 8.

## 5. Discussion

This paper uses an inductive path to explore the measurement dimensions of PSM that fit the college and university context, adding two new dimensions: collectivist tendency and self-improvement. Our study has strong similarities with the study developed by Kaifeng Yang and Huishan Yang (2021) [18] in terms of the cultural background of the target population, and we can use it as a comparison. Table 9 shows the differences between our research and the existing research intuitively. We placed similar dimensions or items covered by each scale in the same row as much as possible, but some items could not be corresponded to each other due to the differences in context and culture.

There are two main differences between the results of this study and the prior study. First, the dimensions are different; Perry’s four-factor scale has four dimensions, while Yang’s survey of Chinese civil servants group inducted the Sinicization scale adding social recognition and reputation and moral adherence to this scale, which is a generalization in line with the Chinese Confucian culture and institutional context. The dimensions of PSM in this study reflect both the traditional culture of East Asian countries and the characteristics of college students. Although Chinese college students have strong independence, their long-standing culture and school education have enabled them to retain this collectivist idea. Since ancient times, Chinese people have been deeply influenced by the culture of “Confucianism”, which seeks a balance between purpose and effectiveness, moral pursuit, and social practicality, so people with high motivation for public service are to a certain extent motivated by “self-improvement”. This self-interest tendency has been recognized in PSM research, and the rational motivation proposed by Perry and Wise recognizes that there is an element of personal utility maximization in public service, that the public good and private gain are interdependent, and that whether a person voluntarily participates in some public service activity is based on the public and private benefits that may result from that activity. Whether or not a person voluntarily participates in some public good activity is based on the potential public and private gains and losses from that activity [49].

In contrast to existing research on Chinese motivation of public service, this study focused on college students who are about to enter the workforce, rather than on social groups with some work experience. We argue that although they are also deeply influenced by Confucianism, the public service motivation of college students differs slightly from that of social groups in terms of expression. First of all, in the minds of college students, collective honor is higher than social fame. Having lived in a collective environment for a long time, contemporary college students, especially undergraduates, value collective honor and a sense of belonging, and they frequently mentioned “making contributions to their organizations” and “doing this to find a sense of collective belonging” when describing their motivation for public service. Secondly, college students mainly live on campus and have simple interests, so they rarely encounter the struggle between truth and falsehood, good and evil, or the need to sacrifice their interests to achieve the interests of others, and more often than not, they choose between pursuing pleasure and finding a platform to improve and develop themselves as much as possible. Regarding the motivation of being a student leader or participating in volunteer activities, many answers included “to improve my communication and leadership skills” and “to expand my circle of friends”. Most importantly, for a long time, the PSM has been thought to come from altruism and dedication to realizing the public interest. However, among college students, we found that the PSM may also partly come from the pursuit of self-development and value realization, such as ability, knowledge, prestige, etc. More than 80% of the respondents believed that there is no contradiction between the PSM partly including self-interest and public service and even may promote it.

Second, the expressions of each dimension are different, and this paper summarizes the specific descriptions from the interview scripts of college students to better fit the life situations of college students. For example, in attraction to policy making, we added “actively concerned about school policies, activities, and students’ rights” and “actively suggesting improvements to policies and measures”; in commitment to the public interest and self-sacrifice, the meanings of the specific items are similar to those of existing studies. In terms of commitment to the public interest and self-sacrifice, the meanings of the specific questions are similar to those of existing studies, but we use words more in line with the tone of university students, such as “sense of mission” and “responsibility”; in terms of compassion, due to the limited circle of undergraduate life, the interviews did not talk about social welfare and public programs but rather favored serving students and vulnerable groups. In terms of collectivism, “giving priority to collective interests” and “willing to contribute to the collective and the country” is similar to the self-sacrifice dimension of the Perry scale, but they are classified as the fifth dimension in this paper. The reason is that when these two dimensions are mentioned in the interviews, they are always inseparable from the word “collective”, so they can be classified as a separate dimensions to better reflect the collectivist thinking of Chinese college students; in terms of self-improvement, “valuing the sense of achievement and honor in public service” seems to be similar to the “making a difference in society means more to me than personal achievements” of Perry’s scale, but the former is more focused on personal achievement and honor, while the latter is more focused on making contributions to society.

## 6. Conclusions

Our study investigated the initial dimensions of PSM of college students, and we conducted a qualitative study using grounded theory and one-on-one interviews. We conducted six dimensions: “attraction to policy making”, “commitment to the public interest”, “compassion”, “self-sacrifice”, “self-sacrifice”, “collectivist tendency”, and “self-improvement”. By comparing the differences between this study and previous studies, and by analyzing the reasons from traditional culture, group characteristics, and ideological education, this study expanded the cross-cultural and cross-group studies of PSM. Our findings suggest that the motivation for public service can be not only altruism but also can be a result of individuals seeking self-development and value. According to grounded theory, we have to distill theory from a wealth of primary sources, so we used the interview method, which is the most direct method to acquire the most primary sources, and considering the typicality of the interviews, the interviewees involved student cadres, non-student cadres, and students from different colleges. We also used the double-blind method to avoid the interviewees’ self-disguised questions.

The significance of this paper is to expand the scope of research on PSM theory and the applicability of the measurement dimensions. Many countries have explored the localization and group-specific applicability of PSM [44,45,46,48], and the specificity of research objects is the direction of future research on PSM. As an important force for social development, college students have their characteristics and are a large number of people who cannot be ignored. At the same time, research on PSM is no longer limited to the public sector; studies point out that PSM not only affects the structure of public sector employees [60] but also has a driving effect on employees engaged in the secondary (business organizations) and tertiary (nonprofit organizations) sectors, citizens, and students [61]. Pre-employment PSM levels predict attractiveness in the public or private sector [62]. Therefore, it is important to study the measurement dimensions of PSM among college students who are about to enter the workforce.

However, this qualitative research lacks supporting evidence from quantitative studies. The subsequent analysis will expand the sample size and test the reliability and validity of the measurement methods.

## Figures and Tables

**Figure 1 ijerph-19-15084-f001:**
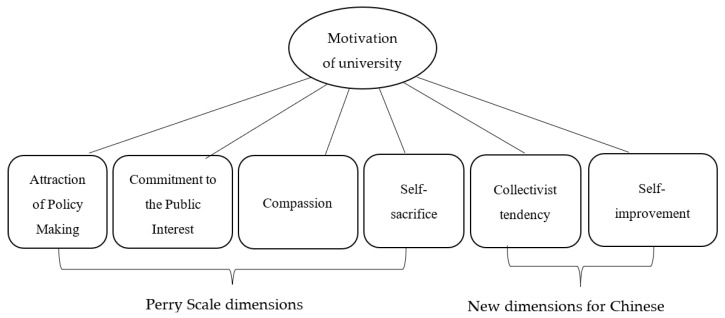
Six dimensions of college students’ motivations for public service.

**Table 1 ijerph-19-15084-t001:** Nodes table of public service motives.

Open Coding	Axial Coding	Selective Coding
Proactive understanding of national events and public social events (18) Proactive attention to school policies, activities, and student rights (32) Take the initiative to improve or make suggestions on policies and measures (23)	Attraction to policy making (73)	Public Service Motivation (592)
Responsible and committed to work (54) Able to take initiative for public service (46) The perceived value of public service to others (24)	Commitment to the public interest (124)	
Focus on the difficulties of vulnerable groups (26) Have a strong sense of helping people (34) Ability to empathize and put yourself in the shoes of others (31)	Compassion (91)	
Willingness to sacrifice personal interests to help others (36) Serving others without regard to reward (14) Dare to take risks and be willing to suffer for the public good (31)	Self-sacrifice (81)	
Prioritizing the collective good over the individual (33) Have a strong sense of collective honor and belonging (22) Happy to contribute to the collective and the country (52) Willingness to emulate the actions of civilian heroes (18)	Collectivist tendency (125)	
Expanding interpersonal relationships (23) Enhancement of personal overall quality (41) Valuing the sense of achievement and honor in public service (19) Perceiving public service as self-fulfillment (15)	Self-improvement (98)	

**Table 2 ijerph-19-15084-t002:** Material information of the sub-node “Attraction to policy making” of axial coding.

Open Coding	Example of Interview Content
Proactive understanding of national events and public social events (18)	*In times of social outbreaks of hardship, many students use the power of the Internet to spread avenues of relief or to promote some knowledge or policy of epidemic prevention, and their motivation for public service is higher compared to those who are not even willing to retweet.*
Proactive attention to school policies, activities, and student rights (32)	*They will pay special attention to some new initiatives of the school and forward news about discipline construction, future development plans of the school, campus competitions, rewards, and punishments in their circle of friends.* *Some students who are involved in clubs or serve as student leaders serve as incentive policymakers and messengers, and they have a high level of interest and involvement in campus activities.*
Take the initiative to improve or make suggestions on policies and measures (23)	*They will take the initiative to go to the college or to the counselor to reflect, and then when there is some lecture information that may not be disclosed very timely, they will also take the initiative to reflect.* *Those who are concerned about the policy will express their views on the policy and give their opinions through the principal’s mailbox.*

**Table 3 ijerph-19-15084-t003:** Material information for the “Commitment to the public interest” sub-node of axial coding.

Open Coding	Example of Interview Content
Responsible and committed to work (54)	*Some people who join a club and then cannot be contacted when they need to discuss and mention ideas, in most cases, don’t want to participate, so they pretend they don’t see it. They are not responsible for the decision to join the club in the first place, and they are not responsible for what they have to do, which is a sign of low public service motivation.*
Able to take initiative for public service (46)	*Students who are highly motivated by public service will actively participate in volunteer activities, club activities, and competition research and are the first to take a stand and participate.* *When the organization encountered difficulties, they were willing to be the first to take up and propose solutions. At that time, there was a job of mailing graduation gifts in the club, and no one was willing to do it, so a junior student took the initiative to step forward, negotiating with the merchants and contacting students, and he was the one doing it.*
The perceived value of public service to others (24)	*For people who go to remote areas or areas where education is relatively backward to teach, I think it is a good manifestation of the value of life and very meaningful.* *When I go to teach and see the children learn something very happily, I will also feel happy and accomplished.*

**Table 4 ijerph-19-15084-t004:** Material information of the sub-node “compassion” by axial coding.

Open Coding	Example of Interview Content
Focus on the difficulties of vulnerable groups (26)	*In their free time, they go to orphanages to provide services and learn sign language or how to read books for the blind so that they can better educate others about the subject.* *Some students in the school have researched the accessibility of the school specifically to facilitate the lives of people with disabilities.*
Have a strong sense of helping people (34)	*XX’s first reaction of righteousness is to rush up, so his original sense of helping others must be very strong, and his subconscious is to go up to stop him.* *I once walked in the street to see a mother and daughter riding a motorcycle with a lot of things, and then things fell quite far—some people are standing next to not helping pick up or pretending not to see, and I was already towards the mother and daughter side, and then help them pick up.*
Ability to empathize and put yourself in the shoes of others (31)	*They may be a little more sensitive to their friends’ points of concern. Perhaps this is also a special observation ability that they have honed in their volunteer or work service, and they will be more likely to catch their friends’ unhappy or frustrating points.* *A senior classmate was given priority for a part-time job at the school because his family was poor, but he allowed a junior classmate, both from the same poor family, but this student cared more about others.*

**Table 5 ijerph-19-15084-t005:** Material information of the “self-sacrifice” sub-node of axial coding.

Open Coding	Example of Interview Content
Willingness to sacrifice personal interests to help others (36)	*During the school’s epidemic prevention period, which required more risky activities such as vaccination and nucleic acid testing, they called on party members to go first, and of course, I would be the first to take time out to redirect myself and then do my best to participate in this task.* *In the event of an epidemic or natural disaster, many students take the initiative to donate, and will also mobilize alumni groups to make donations to their alma mater for epidemic preparedness.*
Serving others without regard to reward (14)	*People who participate in volunteer service don’t care about the extra rewards or subsidies that volunteer hours give them, which is pure selflessness.* *A heart of altruism without regard for personal gain or loss.*
Dare to take risks and be willing to suffer for the public good (31)	*It is quite hard for a transfer student and college student village officials to work at the primary institutions.* *It is great righteousness for those who put their safety behind public safety and choose to bear the harm themselves.*

**Table 6 ijerph-19-15084-t006:** Material information of the sub-node “collectivist tendency” by axial coding.

Open Coding	Example of Interview Content
Prioritizing the collective good over the individual (33)	*They are willing to sacrifice their interests for the sake of the collective good, which I think is already a kind of spiritual state of sacrificing the small for the big.* *The class committee is a thankless job. Whenever something happens, they come forward and when faced with the choice between individual and collective interests, they tend to make the choice that is more beneficial to the collective interests.*
Have a strong sense of collective honor and belonging (22)	*It makes me happy that I can get a sense of belonging when I work and volunteer in the club.* *While balancing individual interests, they will strive for the honor of the group as much as possible.*
Happy to contribute to the collective and the country (52)	*The reason why I chose to stay as a minister is that I want to help the teachers and serve the students.* *When the epidemic was more serious, we contacted local volunteer groups to do what we can.*
Willingness to emulate the actions of civilian heroes (18)	*During the epidemic in Wuhan, party members and medical workers went to the front line, and if they had the opportunity, they would have gone to join their ranks.* *When I saw the heroic deeds in the news, I thought it was a very noble act and wanted to work hard to become this type of person.*

**Table 7 ijerph-19-15084-t007:** Material information of the sub-node “self-improvement” of axial coding.

Open Coding	Example of Interview Content
Expanding interpersonal relationships (23)	*Some people will be very focused on making good relationships with teachers and getting more resources while providing student services.* *In my circle of working as a student, I have met many people and built my circle of university friends.*
Enhancement of personal overall quality (41)	*Serving as a student leader has taught me that the prep for something is very important.* *During my time as a student officer in the Youth League Committee, I have gained a lot of very valuable experience and training opportunities, including communication with people and team coordination.*
Valuing the sense of achievement and honor in public service (19)	*I have participated in many volunteer activities that would give me a great sense of accomplishment.* *Public service has brought me some honorary titles.* *Although I couldn’t get material gains from teaching, I felt the mood that I didn’t have before spiritually and felt it was an honor.*
Perceiving public service as self-worth (15)	*Having an attitude of giving to others, helping others, and hoping to achieve some fulfillment in my life.* *They go to volunteer more to make their lives worthwhile.* *Participation in public service is a form of self-request, proving one’s worth.*

**Table 8 ijerph-19-15084-t008:** Word frequency statistics.

Words	Frequency	Supported Dimensions
Student union	377	APM&CT
Volunteers	222	CPI
Selfless dedication	168	SS
Sense of responsibility	113	API
Communist Party members	101	APM&CT
Be helpful	52	COM
Immortality	37	SI
Sense of values	30	API&SI
Forum	19	APM
Heroism	19	SS
Stand up bravely	18	SS
Put yourself in others’ shoes	13	COM
As possible as they can	6	COM
Same ambition	6	CT

Notes: APM is an abbreviation for attraction to policy making; CPI is an abbreviation for commitment to the public interest; SS is an abbreviation for self-sacrifice; COM is an abbreviation for compassion; CT is an abbreviation for collectivist tendency; SI is an abbreviation for self-improvement.

**Table 9 ijerph-19-15084-t009:** Comparison of PSM measurement dimensions.

	Perry Four Factor Scale	Yang Sinicization Scale	This Article Summarizes the Results
Attraction to policy making	Politics is a dirty word (Reversed)		
	The give and take of public policymakers don’t appeal to me (Reversed)	I would love to be involved in the public policy-making process	Take the initiative to pay attention to school policies, activities, and students’ rights; take initiative to improve or make suggestions on policies and measures
	I don’t care much for politicians (Reversed)	I am happy to share my views on public policy with others	Proactively learn about national events and social public events
		It gives me great satisfaction that others can benefit from the policies I am involved in	
Commitment to the public interest	It is hard for me to get intensely interested in what is going on in my community (Reversed)		
	I unselfishly contribute to my community	I hope that through my efforts I can make a good impact on society	A sense of responsibility and mission for the work
	Meaningful public service is very important to me	Public service that benefits the public is very meaningful to me	
	I would prefer seeing public officials do what is best for the whole community even if it harmed my interests		The perceived value of public service to others
	I consider public service my civic duty	I consider it my civic duty to be actively involved in public service	A proactive approach to public service
Compassion	I am rarely moved by the plight of the underprivileged (Reversed)	I am concerned about the interests of the underprivileged	Focus on the difficulties of vulnerable groups
	Most social programs are too vital to do		
	It is difficult for me to contain my feelings when I see people in distress		Ability to empathize and put yourself in the shoes of others
	To me, patriotism includes seeing to the welfare of others	I always put myself in the shoes of others in life	
	I seldom think about the welfare of people whom I don’t know personally (Reversed)		
	I am often reminded by daily events about how dependent we are on one another		
	I have little compassion for people in need who are unwilling to take the first step to help themselves (Reversed)		Strong sense of helping others
	There are a few public programs that I wholeheartedly support (Reversed)	I am willing to share what I know with others	
Self-sacrifice	Making a difference in society means more to me than personal achievements	For me, making a good change for society is more meaningful than personal achievement	
	I believe in putting duty before self		
	Doing well financially is more important to me than doing good deeds (Reversed)		
	Much of what I do is for a cause bigger than myself		
	Serving citizens would give me a good feeling even if no one paid me for it	I think it is much more important to give than to get	Serving others without regard for reward
	I feel people should give back to society more than they get from it		
	I am one of those few people who would risk a personal loss to help someone else.		Willingness to take risks and suffer for the public good
	I am prepared to make enormous sacrifices for the good of society	I am willing to sacrifice my interests for the public interest	Willing to sacrifice personal interest to help others
Social recognition and reputation		Whenever I think of people who have lived on throughout history, I feel excited and uplifted.	
		I am particularly concerned about whether my behavior conforms to the general norms of society	
		The sense of honor that comes from social recognition is very important to me	
Moral adherence		I think a character is very important to ensure community and social order	
		I will stick to the right things in life, even though I may suffer from the cold eyes of others	
		For the right thing, even if the probability of success is small, we should insist on doing	
Collectivism			Prioritize the collective good over the individual
			Have a strong sense of collective honor and belonging
			Happy to contribute to the collective and the country
			Willingness to emulate the actions of civilian heroes
Self-improvement			Expanding interpersonal relationships
			Improving the overall quality of individuals
			Valuing the sense of achievement and honor in public service
			Believe that public service is the realization of self-worth

## Data Availability

No applicable.

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
