# Peer review of "Analysis of Evaluation Dimensions of Public Service Motivation of Chinese College Students—Qualitative Study Based on Grounded Theory"

_ijerph, 2022, doi:10.3390/ijerph192215084_

Round 1
Reviewer 1 Report
I appreciate the authors for conducting a qualitative study to understand the public service motivation of Chinese college students. However, they are many critical issues that need to be resolved before publication.
1. I failed to understand the meaning of several sentences in the Introduction section. So, I strongly suggest revising the manuscript for English proofreading.
2. I failed to understand the motivation/scope/need for the study
3. Why the "interview" method has been used to understand the public service motivation of Chinese college students? Why not other research methods (Questionnaire, card sorting, etc.)?
4. What type of interview has been conducted in the study?
5. I strongly suggest compressing the "Coding results induction analysis" Section. Also, Separate the "conclusion and discussion" section
6. Overall - I strongly suggest rewriting the full manuscript for better readability and understandability.
Reviewer 2 Report
The authors seek to measure public service motivation in college student groups, using interview and inductive methods to discover new measurement dimensions (collectivist tendencies and self-improvement) based on Perry's research, which is innovative and reflects cultural and national uniqueness. The article is well researched with a solid theoretical analysis. It demonstrates that public service motivation may also originate as a result of an individual's search for self-development and value fulfillment,and has practical implications for the development of college students with high public service motivation at the higher education level. Overall, The conclusions are solid and convicing, and the manuscript is of high quality.
The paper needs to be revised in the following areas: the language expression needs to be unified, especially the use of terms; the research contribution needs to be expressed more clearly; the interpretation of some of the analyses needs to be more explicit; and the literature needs to be updated. The specific issues are as follows:
1. The term "self-improvement," the sixth dimension in the paper, is expressed in different ways, so please standardize the presentation.
2. Please explain on what basis the number of relevant contents/Number of relevant contents/Number of relevant contents are calculated.
3.Please add the literature of the last five years as it is less available.
4.Please highlight what is the importance of the newly discovered dimensions..
5.The section on "Research value and future research directions" that summarizes the article as a whole is proposed to be placed under Section 5.
Round 2
Reviewer 1 Report
I appreciate your corrections. The manuscript is good enough for publication